# Magnolol, A Novel Antagonist of Thrombin and PAR-1, Inhibits Thrombin-Induced Connective Tissue Growth Factor (CTGF) Expression in Vascular Smooth Muscle Cells and Ameliorate Pathogenesis of Restenosis in Rats

**Wen-Chin Ko [1,2]**, **Chia-Ti Tsai [3]**, **Kai-Cheng Hsu [4,5,6]**, **Yu-Che Cheng [2,7,8]**, **Tony Eight Lin [4,5,6]**, **Yi-Ling Chen [1]**, **Chuang-Ye Hong [9]**, **Wan-Jung Lu [10,11]**, **Chun-Ming Shih [12,13,\*]** and **Ting-Lin Yen [14,\*]**

1   Division of Cardiac Electrophysiology, Department of Cardiovascular Center, Cathay General Hospital, 280 Renai Rd. Sec.4, Taipei City 10630, Taiwan; wckcgh@ms17.hinet.net (W.-C.K.); a0925280776@gmail.com (Y.-L.C.)

2   School of Medicine, College of Medicine, Fu Jen Catholic University, 510, Zhongzheng Rd., Xinzhuang Dist., New Taipei City 242062, Taiwan; yccheng@cgh.org.tw

3   Division of Cardiology, Department of Internal Medicine, National Taiwan University Hospital, 7, Chung Shan S. Rd., Zhongzheng Dist., Taipei City 100225, Taiwan; cttsai1999@gmail.com

4   Graduate Institute of Cancer Biology and Drug Discovery, College of Medical Science and Technology, Taipei Medical University, 250 Wu-Hsing Street, Taipei City 11031, Taiwan; piki@tmu.edu.tw (K.-C.H.); tonyelin@tmu.edu.tw (T.E.L.)

5   Ph.D. Program for Cancer Molecular Biology and Drug Discovery, College of Medical Science and Technology, Taipei Medical University, 250 Wu-Hsing Street, Taipei City 11031, Taiwan

6   Ph.D. Program in Biotechnology Research and Development, College of Pharmacy, Taipei Medical University, 250 Wu-Hsing Street, Taipei City 11031, Taiwan

7   Proteomics Laboratory, Department of Medical Research, Cathay General Hospital, 280 Renai Rd. Sec.4, Taipei City 10630, Taiwan

8   Department of Biomedical Sciences and Engineering, National Central University, 300, Zhongda Rd., Zhongli District, Taoyuan City 32001, Taiwan

9   Department of Internal Medicine, Taipei Municipal Wanfang Hospital, 111, Sec. 3, Xinglong Rd., Wenshan Dist., Taipei City 116081, Taiwan; hongprof@tmu.edu.tw

10  Department of Pharmacology, School of Medicine, College of Medicine, Taipei Medical University, 250 Wu-Hsing Street, Taipei City 11031, Taiwan; luwj@tmu.edu.tw

11  Department of Medical Research, Taipei Medical University Hospital, 252 Wuxing St, Xinyi District, Taipei City 11031, Taiwan

12  Department of Internal Medicine, School of Medicine, College of Medicine, Taipei Medical University, 250 Wu-Hsing Street, Taipei City 11031, Taiwan

13  Division of Cardiology, Department of Internal Medicine and Cardiovascular Research Center, Taipei Medical University Hospital, 252 Wu-Hsing Street, Taipei City 11031, Taiwan

14  Department of Medical Research, Cathay General Hospital, 280 Renai Rd. Sec.4, Taipei City 10630, Taiwan

\*   Correspondence: cmshih53@tmu.edu.tw (C.-M.S.); d119096015@tmu.edu.tw (T.-L.Y.); Tel.:+886-2-2737-2181(C.-M.S.); +886-2-8646-1500 (T.-L.Y.)

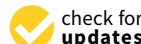

**Featured Application: Treatment of magnolol represents a new therapeutic strategy that may reduce the risk of vascular diseases associated with abnormal vascular smooth muscle cell (VSMC) migration or thrombin/protease-activated receptor 1 (PAR-1) activation.**

**Abstract:** Restenosis and destructive vascular remodeling are the main reasons for treatment failure in patients undergoing percutaneous coronary intervention (PCI). In this study, we explored the efficacy

of magnolol (a traditional Chinese medicine) in the treatment of restenosis. The results of this study showed that the activities of thrombin and PAR-1 (protease-activated receptor 1) were significantly decreased by the treatment of magnolol. Based on protein docking analysis, magnolol exhibits its potential to bind to the PAR-1 active site. In addition, thrombin-induced connective tissue growth factor (CTGF) expression and the upstream of CTGF such as JNK-1 (but not JNK-2), c-Jun, and AP-1 were distinctly inhibited by magnolol (50 μM) in vascular smooth muscle cells (VSMC). For the functional assay, magnolol (50 μM) significantly inhibited the migration of VSMC, and rats treated with magnolol (13 mg/kg/day) after balloon angioplasty has observed a significant reduction in the formation of common arterial neointima. In conclusion, we identified a novel mechanism by which magnolol acts as the thrombin activity inhibitor and may be the PAR-1 antagonist. In accordance with these functions, magnolol could decrease thrombin-induced CTGF expression in VSMCs via PAR-1/JNK-1/AP-1 signaling.

**Keywords:** magnolol; thrombin inhibitor; PAR-1; CTGF; restenosis

## 1. Introduction

Percutaneous coronary intervention (PCI) is a minimally invasive procedure performed to recanalize the coronary arteries in patients with atherosclerosis. Despite the high success rate of PCI, twenty percent of patients have a recurrent major adverse cardiac event in three years [1]. Half of these events are attributive to the initial culprit lesion [1]. So far, restenosis and destructive vascular remodeling remain the major drawbacks of PCI [1,2].

Vascular smooth muscle cell (VSMC) is one of the most important cells in restenosis and vascular remodeling [2]. It switches from a contractile to a synthetic phenotype in response to the injury which increases the stimulatory growth factors or cytokineses such as platelet-derived growth factor (PDGF), interleukin-1, and tumor necrosis factor-α and causes the migration and the proliferation of VSMCs for tissue repair [3,4]. In addition, more than 55% of neointimal hyperplasia cases involve extracellular matrix proteins [5].

Connective tissue growth factor (CTGF), also named CCN2, is a member of the CCN gene family, which comprises six extracellular matrix-associated proteins involved in intercellular signaling. CTGF plays important cellular roles in the proliferation and migration of fibroblasts, the production of the extracellular matrix, and cell adhesion. Therefore, it has related biological effects in wound healing and angiogenesis [6–8]. Previous studies have shown that downregulation of CTGF suppresses neointimal hyperplasia after arterial injury. These suggesting that CTGF may play an important role in-stent restenosis [9].

Thrombin has been reported to be generated abundantly in smooth muscle cells at the site of PCI and amplified by the formation of the prothrombinase complex [10]. Thrombin also mediates several cellular activities involved in the process of restenosis, including smooth muscle cell migration and proliferation [11]. To initiate intracellular signaling, thrombin binds and cleaves the N -terminal domain of the protease-activated receptor (PAR) to expose a new N-terminus.

PARs are a subfamily of G protein-coupled receptors (GPCRs), and its truncated N-terminus acts as a tethered ligand to initiate its own receptor signal [12,13]. PAR-1 mediates cellular responses to thrombin and related proteases [14]. Activated PAR-1 causes platelets to change shape and aggregate and causes phenotypic changes in neurons, smooth muscle cells, immune cells, epithelial cells, and fibroblasts [15]. In normal arteries, PAR-1 expression is detected in platelets, leukocytes, and endothelial cells [16] but is limited in VSMCs. However, after balloon angioplasty insult, PAR-1 transcription is upregulated in VSMCs [17].

Magnolol is a hydroxybiphenyl compound extracted from the root and stem bark of Magnolia officinalis. M. officinalis is an important traditional Chinese and Japanese herbal plant that possesses

numerous medicinal properties and is still used in modern clinical practice [18]. Structural analysis has shown that magnolol is an allosteric potentiator of $GABA_A$ receptors [19]. Artificial magnolol linked dimer is an agonist of peroxisome proliferator-activated receptor, which resides in the nuclear membrane and plays pivotal roles in maintaining lipid and glucose homeostasis [20]. It has also been reported that treating VSMCs cells with magnolol may induce apoptosis via the intrinsic pathway and has the potential to treat restenosis and atherosclerosis [21]. However, the underlying mechanism of magnolol on which thrombin, CTGF expression, and VSMC proliferation-related restenosis remain unclear.

We previously found that thrombin can induce CTGF expression in rat VSMCs and acts on PAR-1 to activate the c-Jun N-terminal kinase (JNK) signaling pathway, which in turn initiates activator protein-1 (AP-1) activation and ultimately induces CTGF expression [22]. This study aimed to investigate the mechanisms of magnolol in thrombin-mediated CTGF expression and in balloon injury-induced neointima formation.

## 2. Materials and Methods

### 2.1. Materials and Reagents

Magnolol (≥98%) was purchased from Chem Faces (Wuhan, Hubei, China). Thrombin (from bovine plasma) and DMSO were purchased from Sigma-Aldrich (St Louis, MO, USA). Dulbecco's modified Eagle's medium (DMEM + GlutaMAX), Lipofectamine 3000 reagent, nonessential amino acids (NEAAs), and fetal calf serum (FCS) were purchased from Thermo Fisher Scientific (Waltham, MA, USA). The penicillin/streptomycin solution was purchased from Biological Industry (Cromwell, CT, USA). For cell viability and cell toxicity, CCK-8 and lactate dehydrogenase (LDH) kit were obtained from Wako (Osaka, Japan). The primary antibody against CTGF was purchased from Abcam (Cambridge, UK). The primary antibody against PAR-1 was purchased from Genetex (Irvine, CA, USA). The cleaved-PAR-1 (Ser42) primary antibody was purchased from Abbkine Scientific (Redlands, CA, USA). The anti-phospho-JNK and anti-JNK antibodies were purchased from Cell Signaling Technology (Danvers, MA, USA). The anti-phospho-c-Jun (pS63) antibody was from Epitomics (Burlingame, CA, USA). The anti-GAPDH antibody was purchased from Thermo Fisher Scientific (Waltham, MA, USA) as an internal control. For immunostaining, CF488A donkey anti-mouse IgG was purchased from Biotium (Fremont, CA, USA). Magnolol was dissolved in DMSO and stored at −20 °C.

### 2.2. Animals

Male Wistar rats (300–350 g) were purchased from BioLASCO (Taipei, Taiwan). All animal experiments and care procedures have been approved by the Institutional Animal Care and Use Committee of Cathay General Hospita (ARC-106-009). Prior to the experimental process, all animals were in a clinically normal state without obvious infection, inflammation, or neurological defects. All animals were separated randomly into three groups: (1) sham, (2) solvent (DMSO), and (3) magnolol (13 mg/kg/day) treated groups. Three animals were used in each group.

### 2.3. Cell Culture

A VSMC line (A10) from the embryonic rat thoracic aorta was purchased from the BCRC (Bioresource Collection and Research Center, Taiwan). The cells were sustained in DMEM-GlutaMAX nutrient mixture containing 10% FCS, 0.1 mM NEAA, 1 mM sodium pyruvate, 100 U/mL penicillin G, and 1 mg/mL streptomycin in a humidified 37 °C incubator with 5% CO2. Cells were then inoculated into 6 cm Petri dishes for western blotting; into 12-well plates for immunofluorescent staining; into 24-well plates for cell viability and cell toxicity assay; into 96-well plates for cell transfection and luciferase assays.

### 2.4. Cell Viability and Cell Toxicity Assay

Cell viability was tested by a colorimetric CCK-8 assay and cell viability was by LDH assay. Briefly, cells ($1.5 \times 10^4$ cells) were cultured in 24-well plates and pre-incubated with the vehicle (DMSO) or various concentrations (10–50 μM) of magnolol for 90 min and then treated with thrombin (1 U) for 24 h. Following the treatment of magnolol and thrombin, CCK-8 (10%) was incubated with the cells for 2 hours for detecting the viability of the cell. The absorbance was measured at 450 nm on a microplate reader. In the LDH assay, the LDH reagent was mixed with the supernatants and incubated for 20 min at room temperature. The absorbance was measured at 550 nm on microplate reader (BioTek Synergy HTX, BioTek Instruments, Winooski, VT, USA). Both of the experiments were repeated at least three times and each experiment was performed in triplicate.

### 2.5. Immunoblotting

A10 cells ($25 \times 10^4$) were serum-starved (0.1% FCS) for 6 h and then incubated with vehicle (DMSO, 0.05%) or magnolol (10–50 μM) for 90 min. After the treatment of magnolol or vehicle, 1 U/mL thrombin was then treated for the indicated time. Subsequently, cells were harvested and homogenized in radioimmunoprecipitation assay (RIPA) lysis buffer with protease and phosphatase inhibitor (Thermo Fisher Scientific, MA, USA). Sodium dodecyl sulfate-polyacrylamide gel electrophoresis (SDS-PAGE) was used to separate the protein sample (loaded with equal amount) via distinct molecular weight and the separated proteins were transferred to polyvinylidene fluoride (PVDF) microporous membrane. Proteins were probed with specific primary antibodies overnight at 4 °C. Enhanced chemiluminescence (ECL) system was used to detect the immune response bands. The image analysis software Image-Pro Plus 4.5. (Media Cybernetics, Maryland, USA) was used to quantify the optical density of the response bands.

### 2.6. Immunofluorescent Staining

A10 cells ($3.3 \times 10^4$) were seeded on cover slides (20 mm) for 24 h. Serum-starved (0.1% FCS) for 6 h and after pre-incubation with vehicle (DMSO, 0.05%) or magnolol (10–50 μM) for 90 min, the cells were treated with thrombin (1 U/mL) for 10 min. Afterward, the A10 cells were washed once with PBS, and then fixed with 4% paraformaldehyde for 10 min. After permeabilizing the cell membrane with 0.1% Triton X-100 for 10 min, 5% BSA in PBS was used to block for another 40 min. The prepared samples were then incubated with primary antibodies overnight at 4 °C. Subsequently, the primary antibodies probed samples were counterstained with 4′,6-diamidino-2-phenylindole (DAPI, 30 μM) and mounted by a mounting buffer (Vector Laboratories, Burlingame, CA, USA). The samples were analyzed and imaged under a Nikon ECLIPSE 80i epifluorescence microscope system using an external mercury lamp light (Model C-SHG1) source (Nikon, Tokyo, Japan).

### 2.7. Thrombin Activity Assay

Thrombin activity was measured by a fluorometric assay kit (K373-100; BioVision, Milpitas, CA, USA). It utilizes the ability of thrombin to proteolytically cleave a synthetic substrate and release a fluorophore, 7-Amino-4-methylcoumarin (AMC), which can be easily quantified by a fluorescence reader. Thrombin (1 U/mL) was mix with magnolol (20 or 50 μM) or vehicle (DMSO, 0.05%) for 3 min and then adjusted to 96 well white polystyrene microplate with buffer containing substrate. After 20 min of reaction, the absorbance was measured at 37 °C (Ex = 350 nm, Em = 450 nm) on a microplate reader (BioTek Synergy HTX, BioTek Instruments, Winooski, VT, USA).

### 2.8. Protein-Ligand Docking

The molecular docking software, LeadIT [23], was used to identify molecular interactions between the compound in the study and PAR1 (PDB ID: 3VW7). The crystal structure was obtained from the RCSB Protein Data Bank [24]. The PAR-1 binding site was prepared using LeadIT. The binding site

was determined as a radius of 12 Å from the co-crystal ligand. Water molecules in the binding site were removed. The docking procedure of LeadIT used the FlexX docking module. A hybrid enthalpy and entropy approach was used. A maximum number of solutions for iteration and fragmentation were set to 200 and 500, respectively. The docking score parameters used default settings.

### 2.9. Transfection and Luciferase Reporter Assays

The AP-1 activity was presented by the AP-1 reporter gene assay with Cignal AP-1 Reporter Assay Kit (Qiagen, Hilden, Germany). In brief, the AP-1-responsive firefly luciferase construct and the constitutively expressing Renilla luciferase construct were transfected with Lipofectamine 3000 reagent (Thermo Fisher Scientific, MA, USA) to A10 cell. Dual-Glo luciferase assay system kit was used to determine the luciferase activity. The luciferase activity was normalized in accordance with the Renilla luciferase activity and quantified as the ratio of the AP-1 activity in the cells.

### 2.10. Wound Healing Assay

A10 cells ($1 \times 10^4$) were seeded on culture-inserts (Ibidi, Regensburg, Germany) and incubated at 37°C. After achieving confluence, sterile forceps were used to gently remove the insert to make a wound gap. The cells were then incubated with vehicle (DMSO, 0.05%) or magnolol (10–50 μM) for 90 min; afterword, all samples were treated with thrombin (1 U/mL). The microscopic images of the cells were captured at 0, 12, and 24 h after treatment of thrombin, and the mean wound widths were used to represent cell migration rates. The images were analyzed by NIS-Elements software.

### 2.11. Animal Model of Balloon Angioplasty

According to the previous study, balloon angioplasty was performed in male Wistar rats (300–350 g) [25]. A 2F embolectomy balloon catheter (Edwards Lifescience, Irvine, CA, USA) was used in this model and all animals were anesthetized with Isoflurane (2.5%) before surgery. Briefly, the balloon of the catheter, which introduced from the external artery (ECA), was inflated to expand the right common carotid artery (CCA). Following this, the balloon was pulled back to cause the injury of the vessel. After three times of the procedure, the catheter was removed and the wound was sutured. The vehicle or magnolol was given to the animal intraperitoneally in the first 3 days. Two weeks (14 days) after the surgery, all animals were euthanized (Isoflurane overdose) and perfused intracardially with PBS. The insulted region of the right CCA was fixed with 4% paraformaldehyde and sliced into 5-μm sections. The biopsies were stained with hematoxylin/eosin and imaged by a digital microscope (Nikon ECLIPSE 80i, Tokyo, Japan). Areas within the external elastic lamina (EEL), the internal elastic lamina (IEL), the lumen area (LA), and the neointimal areas were analyzed using ImageJ 1.53 a software (National Institutes of Health, Bethesda, MD, USA). The formula: (IEL area—lumen area)/(EEL area—IEL area), was used to show the ratio of intima area/media area (I/M) [25].

### 2.12. Statistical Analysis

The data were expressed as the mean ± SD of the results, with the number of observations attached. The normality of the data were first tested using the Kolmogorov–Smirnov test and the analysis of variance (ANOVA) was utilized to compare continuous variables. If the analysis showed a significant difference between the group means, compared each group by the Tukey method. $p < 0.05$ was considered statistically significant.

## 3. Results

### 3.1. The Treatment of Magnolol Will Reduce the Thrombin-Mediated CTGF Expression in VSMCs

We first test the biocompatibility of magnolol in VSMCs to optimize the concentration of magnolol in this study. The VSMCs were treated with various concentrations of magnolol (range from 10 to 50 μM) alone or co-treated with thrombin (1 U/mL). Following this, 24 h later, cells were applied for

a viability test by CCK8 assay (Figure 1A) and LDH release assay (Figure 1B). These results showed that the addition of magnolol alone or with thrombin were harmless to VSMC at the tested concentration.

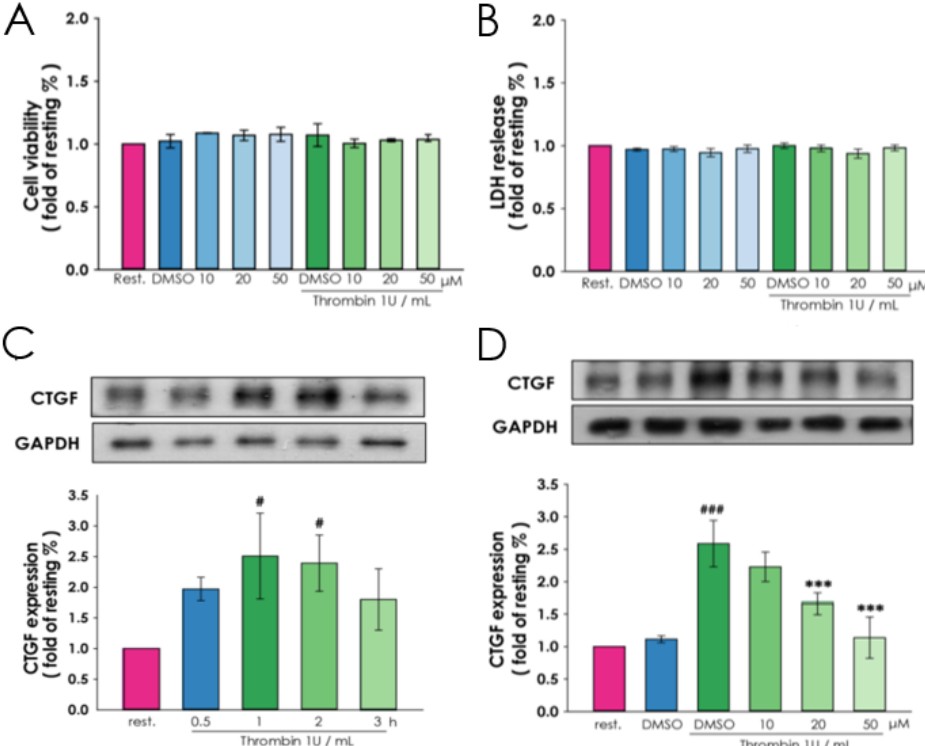

**Figure 1.** Influence of magnolol on thrombin-induced CTGF expression. A10 cells were incubated with magnolol (10, 20 or 50 μM) or vehicle (0.05% DMSO) for 90 min and then treated with or without thrombin (1 U/mL) for 24 h. The cell viability and cytotoxicity were determined by (**A**) CCK-8 assay and (**B**) lactate dehydrogenase (LDH) test, respectively. (**C**) After serum-starved for 6 h, A10 cells were treated with thrombin (1 U/mL) for the indicated times (0.5–3 h), and the protein lysates were collected and subjected to Western blotting for evaluating the time course of CTGF protein expression. (**D**) To assess the influence of magnolol on the thrombin treated cell, the 6-h serum-starved A10 cells were incubated with magnolol (10–50 μM) or vehicle (0.05% DMSO) for 90 min and followed by thrombin (1 U/mL) treatment for 2 h. Data (**A** and **B**) were presented as the means ± SD (n = 3). # $p < 0.05$ and ### $p < 0.0001$, compared with the (**C**) resting group and (**D**) DMSO alone group; *** $p < 0.001$, compared with the (**D**) DMSO alone group. The profiles are all representative examples of three similar experiments.

To reveal the effect of magnolol on thrombin-induced CTGF expression, VSMCs were preincubated respectively with 10, 20 or 50 μM magnolol of magnolol for 90 min prior to the addition of thrombin (1 U/mL) for 1 h. We found that CTGF showed elevated protein expressions after the addition of thrombin. 2 h later, enhanced expression of CTGF reach nearly 2.5 fold than the resting group (Figure 1C) and magnolol significantly reduced the CTGF expression in a dose-dependent manner (Figure 1D). The protein expressions of CTGF were investigated by immunoblotting and normalized with GAPDH. This result demonstrated that magnolol affects to reduce the thrombin-mediated CTGF expression in VSMCs.

### 3.2. Magnolol Hindered the VSMCs from Thrombin-Mediated CTGF Expression via Thrombin Activity Interfering, PAR-1 Cleavage Inhibiting and PAR-1 Active Site Blocking

To understand whether magnolol plays a role in the thrombin-induced CTGF expression, we analyzed the activity of thrombin and the cleavage (activation form) level of PAR-1 after magnolol and thrombin treatment. In Figure 2A, we demonstrated that magnolol (50 μM) significantly down-regulates

the activity of thrombin. Further, we analyzed the level of the cleaved PAR-1 to evaluate the activity of PAR-1. These results showed that the truncated PAR-1 protein was significantly up-regulated 10 min after thrombin treatment (Figure 2B) and magnolol (50 µM) inhibited the production of cleaved PAR-1 protein after thrombin treatment (Figure 2C). According to the results of thrombin activity measurement and cleaved PAR-1 production test, magnolol may have the potential effect as a thrombin activity inhibitor.

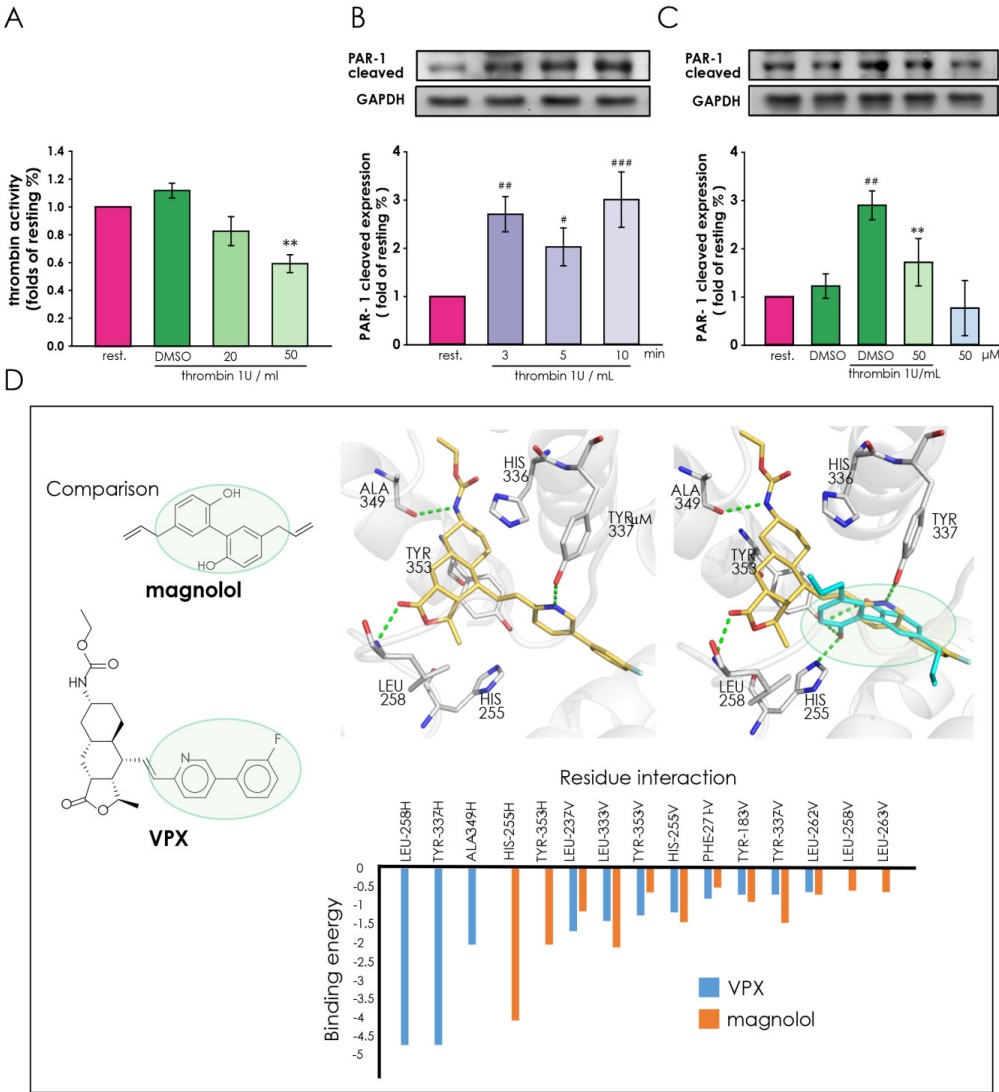

**Figure 2.** The pivotal role of magnolol in thrombin-mediated protease-activated receptor-1 (PAR-1) activation in A10 cells. (**A**) Thrombin activity was measured by a fluorometric assay kit. (**B**) A10 cells were serum-starved for 6 h and then incubated with thrombin (1 U) for the indicated times (0, 3, 5, 10 min) to evaluate the time course of thrombin-induced cleavage of PAR-1. (**C**) Serum-starved A10 cells were incubated with magnolol (10–50 µM) or 0.05% DMSO for 90 min and followed by thrombin (1 U/mL) treatment for 3 min. (**B**,**C**) The cleavage form of PAR-1 was detected through Western blotting under each condition. (**D**) The docking pose of magnolol (top left, blue) in the PAR-1 active site. The docking pose in 2D (top right) highlights both hydrogen and hydrophobic interactions as dotted lines and green lines, respectively. The co-crystal ligand, VPX (bottom, yellow), is superimposed onto magnolol (blue). They contain similar aromatic ring structures that occupy a similar space within the PAR-1 active site (green circle). Green dashed lines denote hydrogen bonds in 3D renderings. Data (**A**–**C**) were presented as the means ± SD (n = 3). ### $p < 0.001$, ## $p < 0.01$ and # $p < 0.05$, compared with the resting group; ** $p < 0.01$, compared with the DMSO plus thrombin treated group.

In addition, we performed a docking analysis to predict the potential of magnolol as a PAR-1 antagonist. Here, we docked magnolol into the PAR-1 crystal structure (PDB ID: 3VW7) and compared its interactions to the co-crystal ligand VPX. Our docking result suggests that magnolol forms interactions within the PAR-1 active site (Figure 3D). Magnolol consists of two chavicol moieties linked together by a carbon-carbon bond between their benzene rings. Two hydrogen bonds, one for each chavicol moiety, are formed between residues H255 and Y353 (Figure 2D). The chavicol moiety consists of a benzene ring and an aliphatic side chain. This facilitates hydrophobic interactions that sandwich the compound in the PAR-1 active site. For example, one chavicol moiety not only forms a hydrogen bond to Y353, but also hydrophobic interactions with residues L237, L263, F271, and L333. The benzene ring of the chavicol moiety also facilitates stable stacking interaction with the phenol ring of Y183. Similar hydrophobic interactions with, residues such as L258, L262, and Y337, sandwich the other chavicol moiety. Then, we compared the binding conformation of magnolol with that of the co-crystal ligand, VPX (Figure 2D). The VPX structure contains two aromatic ring structures linked together (Figure 2D). This is reminiscent of the benzene rings found on magnolol. We found that the aromatic ring structures of VPX occupy a similar space compared to magnolol.

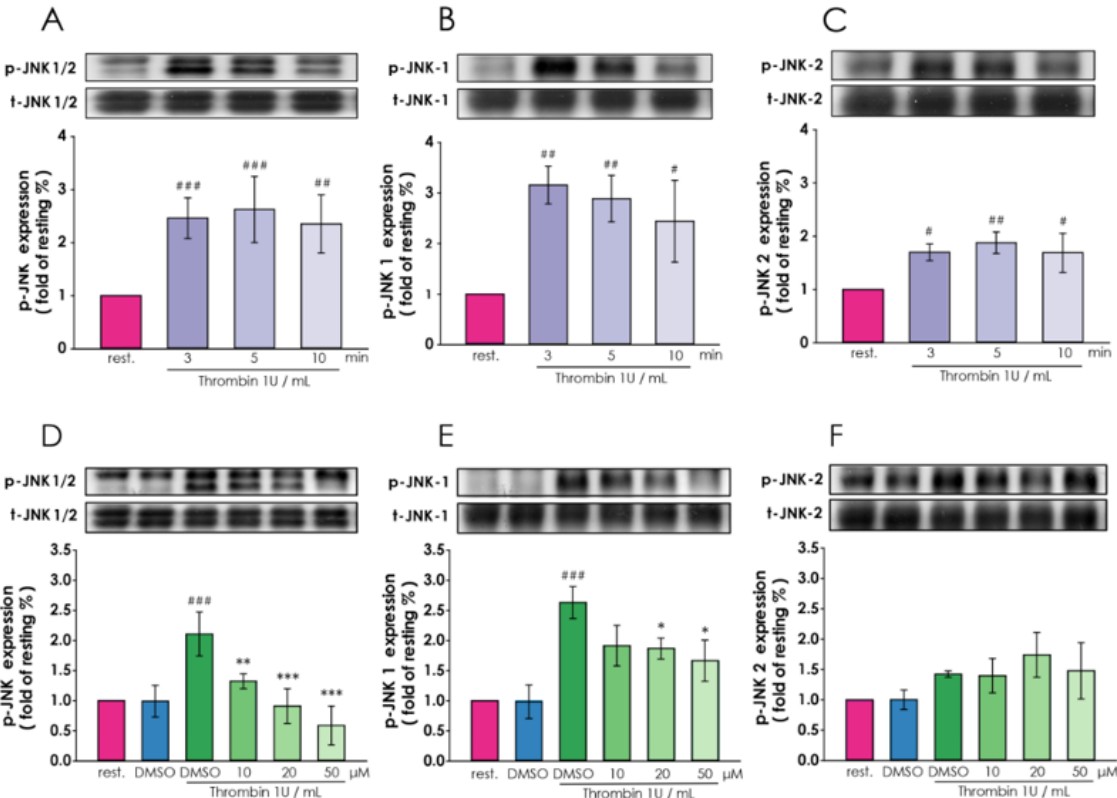

**Figure 3.** Regulatory roles of magnolol in thrombin-induced c-Jun N-terminal kinase (JNK) expression in A10 cells. (**A**–**C**) After 6 h of serum starvation, A10 cells were treated with thrombin (1 U/mL) for 3, 5 or 10 min. The protein lysates were collected and subjected to Western blotting for evaluating the time course of (**A**) JNK1/2, (**B**) JNK-1 and (**C**) JNK-2 phosphorylation that inducted by thrombin. (**D**–**F**) Starved (6 hr) cells pre-treated by magnolol (10–50 μM) or 0.05% DMSO for 90 min and then thrombin (1 U/mL) incubated for 5 min. The (**D**) p-JNK1/2 (**E**) p-JNK-1 and (**F**) p-JNK-2 were detected through Western blotting. (**A**–**C**) # $p < 0.05$, ### $p < 0.001$, ## $p < 0.01$ and # $p < 0.05$, compared with the resting group. (**D**–**F**) ### $p < 0.001$, compared with the DMSO alone group; *** $p < 0.001$, ** $p < 0.01$ and * $p < 0.05$, compared with the DMSO plus thrombin treated group. All data were presented as the means ± standard error of the means (n = 3).

The presence of the hydrophobic interactions sandwiching the aromatic rings of both magnolol and VPX suggests the importance of this feature for potential PAR-1 inhibitors. Further analysis found similar hydrophobic interactions between magnolol, VPX, and PAR-1 residues L237, H255, L262, F271, L333, Y337, and Y353 (Figure 2D). Residue Y183 forms a stable stacking interaction between one of the benzene rings of both compounds. Magnolol forms additional hydrophobic interactions with residues L258 and L263. This is due to the presence of the aliphatic side chains of magnolol. Together, the magnolol can form similar interactions as the ring structures of the co-crystal ligand VPX. This result suggests that magnolol can potentially bind to the PAR1 active site.

### 3.3. Magnolol Performed Different Effect on Thrombin Induced JNK-1 and JNK-2 in Vascular Smooth Muscle Cells

Next, we explored the thrombin-induced expression of JNK-1/2, the upstream molecules of CTGF, under magnolol treatment. After thrombin (1 U/mL) treated for 3, 5, and 10 min, the VSMCs were harvested to analyze the phosphorylation of JNK-1/2. We found that 3 min after 1 U/mL thrombin addition, both P-JNK-1 and P-JNK-2 expression increased (Figure 3A). In Figure 3B,C, these results showed that the protein levels of $p$-JNK-1 and $p$-JNK-2 are 3 and 2.5 times higher than the resting group, respectively. The expressions were digitalized and the $p$-JNK1/2 expressions were normalized with total JNK1/2 and plotted.

Afterward, we pretreated VSMCs with 10, 20 and 50 μM magnolol, and all the cells were harvested 3 min after treatment with 1 U/mL thrombin (Figure 3D). The results showed that compare with vehicle control (DMSO), the expression of $p$-JNK-1 was significantly reduced after pretreatment with magnolol in a dose-dependent manner (Figure 3E). In contrast, magnolol has no significant effect on the expression of $p$-JNK-2 after thrombin treatment (Figure 3F). The expression level of $p$-JNK1 and $p$-JNK2 were digitalized and normalized with the expression of total JNK1/2.

### 3.4. Magnolol Down-Regulate Thrombin-Induced Transcription Factor Activation in Vascular Smooth Muscle Cells

Thrombin is an indicator of c-Jun through the JNK pathway [26,27]. To determine whether magnolol affects the activation of the transcription factor in VSMCs under thrombin treatment, VSMCs were pre-treated with magnolol and harvested after 1 U/mL thrombin treatment. The results indicated that phosphorylated c-Jun showed an elevated level in a time-dependent manner after thrombin treatment (Figure 4A). In Figure 4B, magnolol (10, 20 and 50 μM) inhibit the p-c-Jun level in a concentration-dependent manner. The c-Jun expressions and the phosphorylated level were normalized with GAPDH. As shown in Figure 4C, after 10 min of thrombin (1 U/mL) treatment, phosphorylated c-Jun (red) increased and accumulated significantly in the nucleus (blue). In addition, the red fluorescence intensities were markedly disrupted when the cells were exposed to 50 μM magnolol (Figure 4C, the bottom of the panels).

Next, the VSMCs, that transfected with AP-1-responsive firefly luciferase construct and constitutively expressing Renilla luciferase construct, were used to examine the activating protein-1 (AP-1) activity by AP-1-luciferase reporter gene assay. Transcription factor AP-1, which plays an important role in the regulation of gene expression, cell proliferation, and cell transformation, is mainly composed of Jun (c-Jun, JunB, and JunD) and Fos (cFos, fosB, Fra1, and Fra2) protein families [28,29]. In addition, phosphorylation of AP-1 proteins such as c-Jun probably enhances the transcriptional activity of AP-1 [30]. We found that the AP-1 transactivation activity increased significantly compared with the solvent control group after incubation with thrombin (Figure 4D, lane 3), and the up-regulated AP-1 transactivation activity was reversed by magnolol treatment (Figure 4D, lane 4). These results suggest that magnolol plays an essential role in thrombin-induced c-Jun activity in smooth muscle cells.

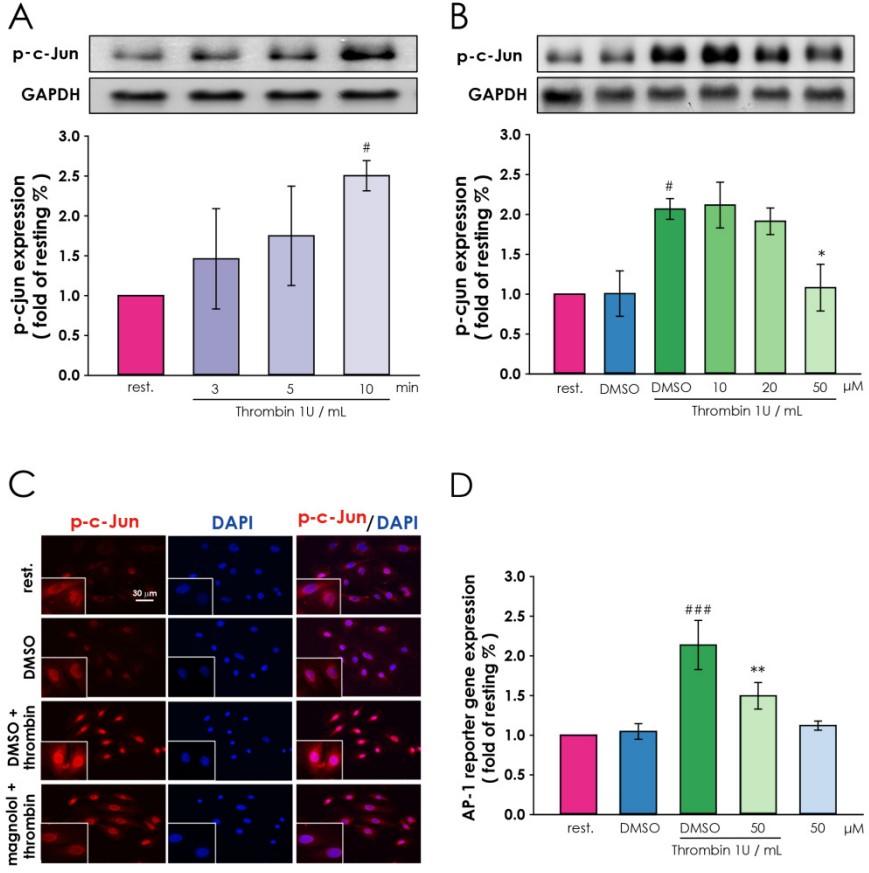

**Figure 4.** Interference by magnolol in thrombin-induced downstream signal transduction, including c-Jun and AP-1 in A10 cells. (**A**) 6 h after serum-starvation, A10 cells were treated with thrombin (1 U/mL) to assess the time course of thrombin-induced c-Jun phosphorylation (**B**) 6 h serum-starved A10 cells were incubated with magnolol (10–50 μM) or 0.05% DMSO for 90 min and then thrombin (1 U/mL) treatment for 10 min. Western blotting was then used to determine the level of p-c-Jun under each condition. (**C**) The location of p-c-Jun (red) was probed using an anti-p-c-Jun antibody, and DAPI (blue) was used to stain the nucleus; scale bar = 30 μm. (**D**) Cells were transiently transfected with 100 ng of AP-1-responsive firefly luciferase construct and constitutively expressing Renilla luciferase construct plasmid for 24 h and then treated with 6 h serum-starvation. Afterward, magnolol (50 μM) or 0.05% DMSO was treated for 90 min and followed by thrombin (1 U/mL) treatment for 16 h. Cells were harvested for the luciferase activity assay. # $p < 0.05$, compared with the (**A**) resting group and (**B**,**D**) DMSO alone group; ### $p < 0.001$, compared with the DMSO alone group in (**D**); * $p < 0.05$ and ** $p < 0.01$, compared with the (**B**,**D**) DMSO plus thrombin treated group. All data were presented as the means ± SD (n = 3). The profile (**C**) was representative examples of three similar experiments.

### 3.5. Treatment of Magnolol Moderates the Migration of VSMCs and Mitigates Restenosis

To evaluate the effect of magnolol on restenosis, we examined thrombin-induced cell migration in vitro and balloon angioplasty-induced neointimal formation in vivo. Migration of VSMCs is a key element in atherosclerosis and restenosis [31]. Figure 5A,B demonstrated that the migration rate was 2 folds higher than the resting group and reversed by the treatment of magnolol (50 μM). The results of in vivo studies (Figure 5C) shown that compared with the sham-operated group, DMSO-treated rats have significantly increased neointimal formation in the balloon-injured common carotid artery. Taking magnolol (13 mg/kg/day) showed a significant decrease in the I/M ratio compared to the DMSO-treated group (Figure 5D).

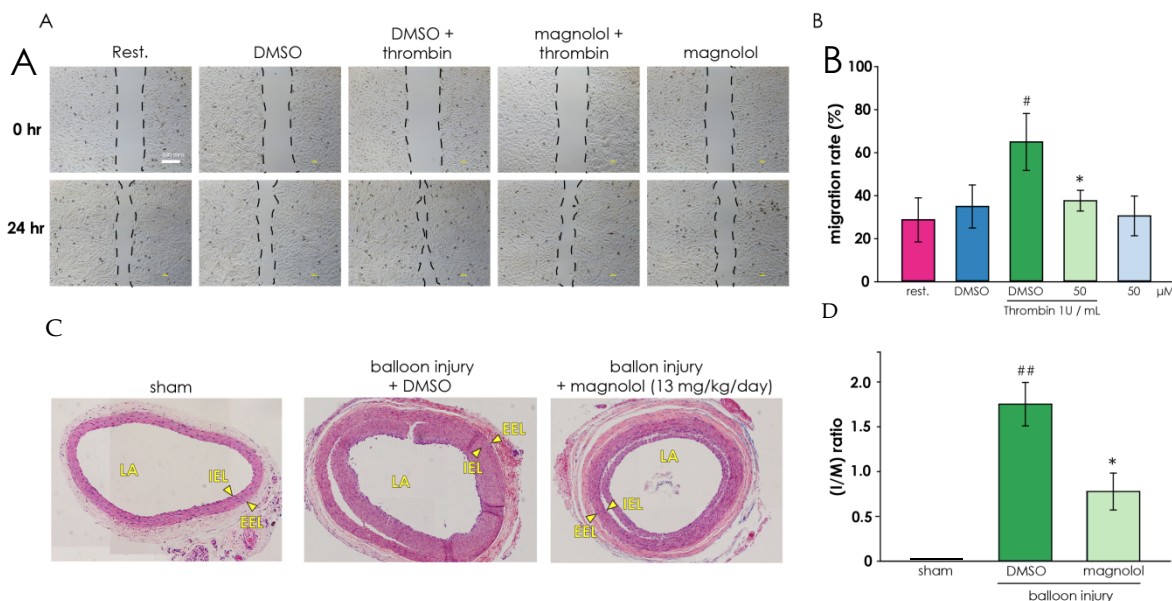

**Figure 5.** Magnolol inhibits the migratory potential of A10 cells and mitigates restenosis. (**A**) The effect of magnolol treatment on the migratory potential of A10 cells was analyzed through the wound healing assay. Representative photomicrographs of initial and final wounds were shown at 40 × magnification and representative examples of three similar experiments. (**B**) Cell migration rate data were presented as the means ± SD of the means (n = 3). # $p < 0.05$, compared with the DMSO alone group; * $p < 0.05$, compared with the DMSO plus thrombin treated group. (**C**) Effects of magnolol on balloon angioplasty-induced neointimal formation in rat common carotid artery. Each experiment group (sham operation group, solvent control group (DMSO), and magnolol (13 mg/kg/day) treated group) were taken microscopic images 14 days after balloon angioplasty. The compiled statistics are shown in (**D**). Data are expressed as mean ± standard deviation (*n* = 3). ## $p < 0.01$, compared with the sham group; * $p < 0.05$, compared to the solvent (DMSO) group. LA, lumen area; IEL, internal elastic lamina; EEL, external elastic lamina.

## 4. Discussions

This study provides the first evidence for a novel mechanism by which magnolol, a widely used traditional medicine, inhibits the activity of thrombin, activation of PAR-1, and subsequently downregulates CTGF expression. We also demonstrated that magnolol attenuated thrombin-induced smooth muscle migration in vitro and balloon angioplasty -induced neointimal formation in vivo.

Thrombin-mediated activation of PAR-1 has been reported to have numerous biological effects. Therefore, preventing PAR-1 activation (cleavage) and blocking the activation site of PAR-1 is very important for preventing subsequent cellular responses of PAR-1. This study demonstrated that magnolol (50 μM) significantly reduced the activity of thrombin (Figure 2A) and decreased thrombin-regulated PAR-1 cleavage (Figure 2C) in VSMCs. We also used protein-ligand docking analysis to indicate that magnolol potentially binds to the activation site of PAR-1. These results suggest that magnolol is a thrombin inhibitor and a PAR-1 antagonist. Therefore, treatment with magnolol may be a beneficial strategy for thrombin/PAR-1-related diseases such as thrombosis formation [32], tumor progression [33], and arterial restenosis [34].

In previous studies, JNK-1 and JNK-2 isoforms have been indicated to exhibit different functions in obesity, insulin resistance, axonal guidance, etc. [35,36]. In this study, we showed that magnolol specifically downregulated the phosphorylation degree of JNK-1 upon thrombin treatment (Figure 3D–F). JNK-1 has been reported to be the major c-Jun/AP-1 interacting kinase after cells are stimulated by thrombin [37,38]. We demonstrated that thrombin-induced activation of JNK-1 (Figure 3B), c-Jun (Figure 4A) and AP-1 (Figure 4C,D) in VSMCs. In addition, thrombin-stimulated p-JNK-1 (Figure 3D–F), p-c-Jun (Figure 4B) and AP-1 (Figure 4C,D) were inactivated by magnolol

administration. According to a previous study, balloon angioplasty-induced smooth muscle cell proliferation is mediated by the rapid activation of JNK-1 [39]. Moreover, it has been reported that cell proliferation of hepatocytes, tumor cells, and endothelial cells is related to JNK-1 but not to the activation of JNK-2 [40–42]. Based on this evidence, magnolol has the potential to ameliorate VSMC proliferation through specific JNK-1 inhibition and to treat diseases such as cancer and atherosclerosis. However, it is worth noting that JNK-1 deficiency in hematopoietic cells may increase the incidence of atherosclerosis [43].

In our previous work, we found that thrombin induces the expression of CTGF in the rat VSMCs through PAR-1/JNK/AP-1 pathway, suggesting that CTGF may involve in the pathogenesis of in-stent restenosis and atherosclerosis [22]. In this study, we demonstrated that treatment with magnolol reduced thrombin-induced CTGF expression (Figure 1C) via the PAR-1/JNK-1/AP-1 signaling cascade. In addition, VSMC proliferation (Figure 5A) and vascular restenosis caused by balloon angioplasty (Figure 5C) were ameliorated by magnolol treatment. It has been shown that CTGF contributes significantly to the proliferation of lung fibroblasts, adipose-derived stromal cells, and mesenchymal stromal cells [44–46]. CTGF also mentioned that it participates in the occurrence and development of restenosis through paracrine action, and overexpression of CTGF may weaken the stability of advanced atherosclerotic plaques [47]. The inhibition of CTGF expression improves cardiac repair and suppresses fibrosis after myocardial infarction [48]. Therefore, magnolol may be a new therapeutic agent for complications caused by myocardial fibrosis and CTGF-related diseases.

Even though we focused on the role of magnolol in the PAR-1/JNK/AP-1 pathway, there were several mechanisms that should not be neglected in research regarding smooth muscle cell activation and restenosis. Magnolol has been reported to be an inhibitor of signal transducer and activator of transcription protein 3 (STAT3) [49]. Thrombin activates STAT3 via PAR-1 pathway [50], and the STAT3 activation is an important molecular mechanism for restenosis and smooth muscle cell activation [51]. Furthermore, thrombin activated nuclear factor kappa-light-chain-enhancer of activated B cells (NF-κB) was reported to involve in the mechanism of VSMCs proliferation, atherosclerosis, and restenosis [52]. An inhibitor, such as magnolol, of NF-κB, has shown the ability to reduce the VSMCs proliferation and benefit restenosis [53].

In this study, we revealed a novel function of magnolol as a thrombin inhibitor and PAR-1 antagonist. Currently, vorapaxar is the only approved therapeutic agent targeting PAR-1 [54], but due to safety considerations, its clinical application is restricted [15]. According to the adoption of new strategies, many new reagents have been designed to circumvent the limitations of the previously developed PAR-1 modulation [15]. On the other hand, thrombin as an activator of PAR-1, inhibiting thrombin activity may be an alternative plan to regulate PAR-1 activity. Oral direct thrombin inhibitors (DTIs) such as Dabigatran etexilate has considered to modulating the activity of PAR-1 many advantages, such as good safety and wide treatment range [55]. Magnolol is an important traditional Chinese and Japanese herbal plant and is still used in modern clinical practice. Similar to Varapraxa and dabigatran etexilate, magnolol may block the active site of PAR-1 (Figure 2D) and has the ability to inhibit thrombin activity (Figure 2A). Most importantly, magnolol did not harm VSMC in our study (Figure 1A,B). Therefore, magnolol has the potential as a new therapeutic agent against thrombin and PAR-1 related diseases.

Inhibition of PAR-1 activity on platelets will affect the platelets aggregation, thereby causing the possibility of bleeding–particularly intracranial hemorrhage [56]. Previous studies have pointed out that Vorapaxary is at risk of bleeding in patients with certain diseases (e.g., diabetics) or concomitant use of other anticoagulants (such as aspirin, clopidogrel) [15]. However, in patients with PCI and CABG, the use of Vorapaxar does not significantly increase the side effects of bleeding [15]. Compared with Vorapaxar, magnolol has a variety of biological activities, such as anti-inflammatory, anti-free radical and neuroprotective ability [57]. In patients with cardiovascular diseases, it may have a better protective effect than Vorapaxar due to its multiple effects. In order to avoid bleeding by systemic effects, local administration of magnolol (for example, a drug-eluting balloon) may be considered.

In addition, the applicable indications of Magnolol, the applicable blood concentration and whether it can be co-treated with other antiplatelet drugs are worthy of further research in the future.

In conclusion, our results provide a mechanism that links the inactivation of thrombin and PAR-1 caused by magnolol with magnolol-elicited JNK-1, c-Jun, and AP-1 inactivation, which ultimately reduces CTGF expression in VSMCs. These findings may provide insights into the treatment strategies of the pathogenesis of in-stent restenosis and offer a variety of medical options for patients with thrombin/PAR-1 related diseases.

**Author Contributions:** Conceptualization, W.-C.K., C.-Y.H., C.-M.S., C.-T.T., W.-J.L. and T.-L.Y.; software, K.-C.H. and T.E.L.; formal analysis, Y.-L.C. and T.-L.Y.; investigation, W.-C.K., Y.-L.C. and T.-L.Y.; resources, W.-C.K. and T.-L.Y.; data curation, Y.-L.C. and T.-L.Y.; writing—original draft preparation, T.-L.Y., Y.-C.C. and W.-C.K.; writing—T.-L.Y. and W.-C.K.; visualization, Y.-L.C. and T.-L.Y.; supervision, T.-L.Y. and W.-C.K.; project administration, T.-L.Y. and W.-C.K. All authors have read and agreed to the published version of the manuscript.

**Funding:** This work was funded by Cathay General Hospital (95CGH-TMU-08 and CGH-MR-A10602).

**Acknowledgments:** Authors thank Jung San Huang for editing the manuscript.

**Conflicts of Interest:** The authors declare no conflict of interest.

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
