# Peer review of "Magnolol, A Novel Antagonist of Thrombin and PAR-1, Inhibits Thrombin-Induced Connective Tissue Growth Factor (CTGF) Expression in Vascular Smooth Muscle Cells and Ameliorate Pathogenesis of Restenosis in Rats"

_applsci, doi:10.3390/app10238729_

Round 1

Reviewer 1 Report

Ko et al. submitted an interesting paper concerning magnolol's influence on thrombin-induced CTGF expression in vascular smooth muscle cells and its role in the restenosis process. The authors found magnolol can inhibit thrombin-dependent expression of the JNK signaling pathway and AP-1 transcription factor action, thus downregulating SMC pathological activation. The presented results support the conclusions mentioned above. However, it should be noted that the thrombin-induced JNK signaling pathway regulates the AP-1 transcription factor but, more importantly, modulates STATs activity, especially STAT3 protein (J Neuroimmunol 2008; 204: 118-25). STAT3 activation is an essential molecular mechanism of restenosis and smooth muscle cell activation (Ex. J Cell Mol Med 2013, 17:989-1005). Also, magnolol is a know STAT3 inhibitor (Br J Pharmacol 2006; 148:226-32). Thus, I strongly recommend verifying whether, in the analyzed experimental design in the A10 cell line, magnolol action is also mediated via the STAT3 pathway. Further, regardless of AP-1, also NF-kB is an important factor in thrombin action. Please, also address this issue in your experiments and discussion.

Author Response

Dear reviewer, thank you for the comment. We have responded to your comment and please see the attachment.

Reviewer 2 Report

This article intended to explore further mechanism downward pathway of Magnolol

  1. This study focus on PAR-1, however, there are PAR-3, -4 which also being activated by thrombin. From your previous research article (ref 22), PAR-4 has been demonstrated of less effect. Can Magnolol activate through PAR-3 pathway as well?
  2. How can the authors be sure that the activation is through vascular smooth muscle cells instead of platelets pathway which is also active by thrombin in animal model? Because decreasing platelet activation also would reduce restenosis.
  3. Though animal studies had been performed, the number was small, only 3 in each. The authors might add Vorapaxar which is PAR-1 inhibitor compared with Magnolol.
  4. Can the authors clarify the different between the roles of Magnolol and Vorapaxar which all affect PAR-1 pathway in discussion section. Why is Magnolol promising in treatment of coronary artery disease while Vorapaxar failed due to bleeding?

Author Response

Dear reviewer, thank you for the comment. We have responded to your comments and please see the attachment.

Round 2

Reviewer 1 Report

I understand that the authors were not able to perform additional experiments. The updated discussion explains the other possible signaling pathways, that could be engaged in magnolol action. It would be interesting to verify their significance in future studies.